# DepthSplat: Connecting Gaussian Splatting and Depth

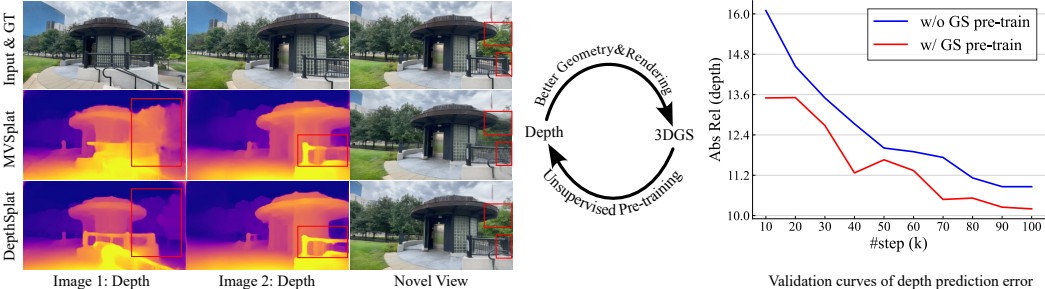

Figure 1: **DepthSplat enables cross-task interactions between Gaussian splatting and depth**. Left: Better depth leads to improved Gaussian splatting reconstruction. Right: Unsupervised depth pre-training with Gaussian splatting leads to reduced depth prediction error.

## ABSTRACT

Gaussian splatting and single/multi-view depth estimation are typically studied in isolation. In this paper, we present DepthSplat to connect Gaussian splatting and depth estimation and study their interactions. More specifically, we first contribute a robust multi-view depth model by leveraging pre-trained monocular depth features, leading to high-quality feed-forward 3D Gaussian splatting reconstructions. We also show that Gaussian splatting can serve as an unsupervised pre-training objective for learning powerful depth models from large-scale unlabelled datasets. We validate the synergy between Gaussian splatting and depth estimation through extensive ablation and cross-task transfer experiments. Our DepthSplat achieves state-of-the-art performance on ScanNet, RealEstate10K and DL3DV datasets in terms of both depth estimation and novel view synthesis, demonstrating the mutual benefits of connecting both tasks. We invite the readers to view our supplementary video for feed-forward reconstruction results of large-scale or 360 scenes. Our code and models will be publicly available.

## 1 INTRODUCTION

Novel view synthesis (Buehler et al., 2001; Zhou et al., 2018) and depth prediction (Schönberger et al., 2016; Eigen et al., 2014) are two fundamental tasks in computer vision, serving as the driving force behind numerous applications ranging from augmented reality to robotics and autonomous driving. There have been notable advancements in both areas recently.

For novel view synthesis, 3D Gaussian Splatting (3DGS) (Kerbl et al., 2023) has emerged as a popular technique due to its impressive real-time performance while attaining high visual fidelity. Recently, advances in feed-forward 3DGS models (Charatan et al., 2024a; Chen et al., 2024; Szymanowicz et al., 2024) have been introduced to alleviate the need for tedious per-scene optimization, also enabling few-view 3D reconstruction. The state-of-the-art sparse-view method MVSplat (Chen et al., 2024) relies on feature matching-based multi-view depth estimation (Xu et al., 2023) to localize the 3D Gaussian positions, which makes it suffer from similar limitations (*e.g.*, occlusions, texture-less regions, and reflective surfaces) as other multi-view depth methods (Schönberger et al., 2016; Yao et al., 2018; Gu et al., 2020; Wang et al., 2021; Duzceker et al., 2021).

On the other hand, significant progress has been made in monocular depth estimation, with recent models (Yang et al., 2024a; Ke et al., 2024; Yin et al., 2023; Fu et al., 2024; Ranftl et al., 2020;

Eftekhar et al., 2021) achieving robust predictions on diverse in-the-wild data. However, these depths typically lack consistency across views, constraining their performance in downstream tasks (Wang et al., 2023; Yin et al., 2022). In addition, both state-of-the-art multi-view (Cao et al., 2022; Xu et al., 2023) and monocular (Yang et al., 2024a; Piccinelli et al., 2024; Yang et al., 2024b) depth models are trained with ground truth depth supervision, which prevents exploiting large unlabelled datasets for more robust depth predictions.

The integration of 3DGS with single/multi-view depth estimation presents a compelling solution to overcome the individual limitations of each technique while at the same time enhancing their strengths. To this end, we introduce *DepthSplat*, which exploits the complementary nature of sparse-view feed-forward 3DGS and robust single/multi-view depth estimation to improve the performance for both tasks.

Specifically, we first contribute a robust multi-view depth model by integrating pre-trained monocular depth features (Yang et al., 2024b) to the multi-view feature matching branch, which not only maintains the consistency of multi-view depth models but also leads to more robust results in situations that are hard to match (*e.g.*, occlusions, texture-less regions and reflective surfaces). The predicted multi-view depth maps are then unprojected to 3D as the Gaussian centers, and we use an additional lightweight network to predict other remaining Gaussian parameters. They are combined together to achieve novel view synthesis with the splatting operation (Kerbl et al., 2023).

Thanks to our improved multi-view depth model, the quality of novel view synthesis with Gaussian splatting is also significantly enhanced (see Fig. 1 left). In addition, our Gaussian splatting module is fully differentiable, which requires only photometric supervision to optimize all model components. This provides a new, unsupervised way to pre-train depth prediction models on large-scale unlabeled datasets without requiring ground truth geometry information. The pre-trained depth model can be further fine-tuned for specific depth tasks and achieves superior results over training from scratch (see Fig. 1 right, where unsupervised pre-training leads to improved performance).

We conduct extensive experiments on the large-scale TartanAir (Wang et al., 2020), ScanNet (Dai et al., 2017) and RealEstate10K (Zhou et al., 2018) datasets for depth estimation and Gaussian splatting tasks, as well as the recently introduced DL3DV (Ling et al., 2023) dataset, which features complex real-world scenes and thus is more challenging. Under various evaluation settings, our DepthSplat achieves state-of-the-art results. The strong performance on both tasks demonstrates the mutual benefits of connecting Gaussian splatting and depth estimation.

## 2 RELATED WORK

**Multi-View Depth Estimation**. As a core component of classical multi-view stereo pipelines (Schönberger et al., 2016), multi-view depth estimation exploits multi-view photometric consistency across multiple images to perform feature matching and predict the depth map of the reference image. Recently, many learning-based methods (Yao et al., 2018; Gu et al., 2020; Wang et al., 2021; Duzceker et al., 2021; Wang et al., 2022; Ding et al., 2022; Cao et al., 2022) have been proposed to improve depth accuracy. For example, MVSNet (Yao et al., 2018) uses the plane-sweep algorithm (Collins, 1996) to build a 3D cost volume, regularizes it with a 3D CNN, and then regresses the depth map. Though these learning-based methods significantly improve the depth quality when compared to traditional methods (Galliani et al., 2015; Schönberger et al., 2016; Xu & Tao, 2019), they cannot handle challenging situations where the multi-view photometric consistency assumption is not guaranteed, *e.g.*, occlusions, low-textured areas, and non-Lambertian surfaces.

**Monocular Depth Estimation**. Recently, we have witnessed significant progress in depth estimation from a single image (Ranftl et al., 2020; Bhat et al., 2023; Yin et al., 2023; Yang et al., 2024a; Ke et al., 2024), and existing methods can produce surprisingly accurate results on diverse in-the-wild data. However, monocular depth methods inherently suffer from scale ambiguities and are not able to produce multi-view consistent depth predictions, which are crucial for downstream tasks like 3D reconstruction (Yin et al., 2022) and video depth estimation (Wang et al., 2023). In this paper, we leverage the powerful features from a pre-trained monocular depth model (Yang et al., 2024b) to augment feature-matching based multi-view depth estimation, which not only maintains high multi-view consistency but also leads to significantly improved robustness in challenging situations such as low-textured regions and reflective surfaces.

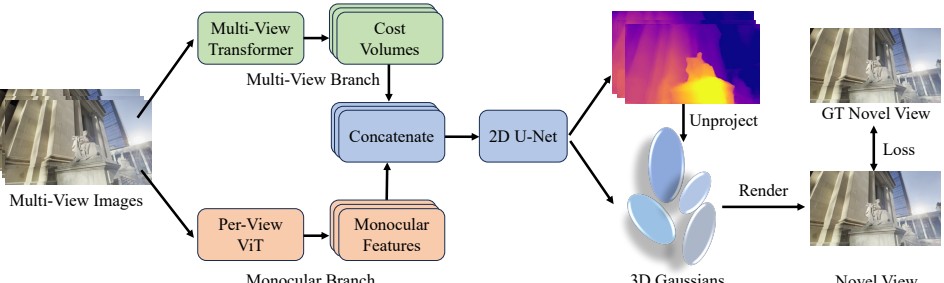

Figure 2: **DepthSplat** connects depth estimation and 3D Gaussian splatting with a shared architecture, which enables cross-task transfer. In particular, DepthSplat consists of a multi-view branch to model feature-matching information and a single-view branch to extract monocular features. The per-view cost volumes and monocular features are concatenated for depth regression with a 2D U-Net architecture. For the depth estimation task, we train the depth model with ground truth depth supervision. For the Gaussian splatting task, we first unproject all depth maps to 3D as the Gaussian centers, and in parallel, we use an additional head to predict the remaining Gaussian parameters. Novel views are rendered with the splatting operation. The full model for novel view synthesis is trained with the photometric rendering loss, which can also be used as an unsupervised pre-training stage for the depth model.

**Monocular and Multi-View Depth Fusion**. Several previous methods (Bae et al., 2022; Li et al., 2023; Yang et al., 2022; Cheng et al., 2024) try to fuse monocular and multi-view depths to alleviate the limitations of feature matching-based multi-view depth estimations. However, they either fuse single- and multi-view depth predictions with additional networks or rely on sophisticated architectures. In contrast, we identify the power of off-the-shell pre-trained monocular depth models and propose to augment multi-view cost volumes with monocular features, which not only leads to a simple model architecture but also strong performance.

**Feed-Forward Gaussian Splatting**. Several feed-forward 3D Gaussian splatting models (Charatan et al., 2024a; Szymanowicz et al., 2024; Chen et al., 2024; Wewer et al., 2024; Tang et al., 2024; Xu et al., 2024; Zhang et al., 2024) have been proposed in literature thanks to its efficiency and ability to handle sparse views. In particular, pixelSplat (Charatan et al., 2024a) and Splatter Image (Szymanowicz et al., 2024) predict 3D Gaussians from image features, while MVSplat (Chen et al., 2024) encodes the feature matching information with cost volumes and achieves better geometry. However, it inherently suffers from the limitation of feature matching in challenging situations like texture-less regions and reflective surfaces. In this paper, we propose to integrate monocular features from pre-trained monocular depth models for more robust depth prediction and Gaussian splatting reconstruction. We additionally study the interactions between depth and Gaussian splatting tasks. Another line of work like LGM (Tang et al., 2024), GRM (Xu et al., 2024), and GS-LRM (Zhang et al., 2024) relies significantly on the training data and compute, discarding explicit feature matching cues and learning everything purely from data. This makes them expensive to train (*e.g.*, GS-LRM (Zhang et al., 2024) is trained with 64 A100 GPUs for 2 days), while our model can be trained in 1 day with 8 GPUs. Moreover, our Gaussian splatting module, in return, enables pre-training depth model from large-scale unlabelled datasets without the need for ground truth depths.

**Depth and Gaussian Splatting**. In addition to the experimental setup (feed-forward) studied in this paper, recently, there has been another line of work (Chung et al., 2024; Turkulainen et al., 2024) which applies an additional depth loss in the Gaussian splatting optimization process. We remark that these two approaches (feed-forward *vs.* per-scene optimization) are orthogonal. In particular, our approach focuses on exploring the advanced network architectures and the power of large-scale training data, while the optimization methods mainly study the effect of loss functions for regularizing the optimization process.

## 3 DEPTHSPLAT

Given $N$ input images $\{\boldsymbol{I}^i\}_{i=1}^N$, ($\boldsymbol{I}^i \in \mathbb{R}^{H \times W \times 3}$, where $H$ and $W$ are the image sizes) with corresponding projection matrices $\{\mathbf{P}_i\}_{i=1}^N$, ($\mathbf{P}_i \in \mathbb{R}^{3 \times 4}$, computed from the intrinsic and extrinsic

matrices), our goal is to predict dense per-pixel depth $\mathbf{D}_i \in \mathbb{R}^{H \times W}$ and per-pixel Gaussian parameters $\{(\boldsymbol{\mu}_j, \alpha_j, \boldsymbol{\Sigma}_j, \boldsymbol{c}_j)\}_{j=1}^{H \times W \times N}$ for each image, where $\boldsymbol{\mu}_j$, $\alpha_j$, $\boldsymbol{\Sigma}_j$ and $\boldsymbol{c}_j$ are the 3D Gaussian's position, opacity, covariance, and color information. As shown in Fig. 2, the core of our method is a multi-view depth model augmented with monocular depth features, where we obtain the position $\boldsymbol{\mu}_j$ of each Gaussian by unprojecting depth to 3D with camera parameters, and other Gaussian parameters are predicted by an additional lightweight head.

More specifically, our depth model consists of two branches: one for modeling feature matching information using cost volumes, and another for extracting monocular features from a pre-trained monocular depth network. The cost volumes and monocular features are concatenated together for subsequent depth regression with a 2D U-Net and a softmax layer. For the depth task, we train our depth model with ground truth depth supervision. Our full model for novel view synthesis is trained with the photometric rendering loss, which can also be used as an unsupervised pre-training stage for the depth model. In the following, we introduce the individual components.

### 3.1 MULTI-VIEW FEATURE MATCHING

In this branch, we extract multi-view features with a multi-view Transformer architecture and then build multiple cost volumes that correspond to each input view.

**Multi-View Feature Extraction**. For $N$ input images, we first use a lightweight weight-sharing ResNet (He et al., 2016) architecture to get $s\times$ downsampled features for each image independently. To handle different image resolutions, we make the downsampling factor $s$ flexible by controlling the number of stride-2 $3 \times 3$ convolutions. For example, the downsampling factor $s$ is 4 if two stride-2 convolutions are used and 8 if three are used. To exchange information across different views, we use a multi-view Swin Transformer (Liu et al., 2021; Xu et al., 2022; 2023) which contains six stacked self- and cross-attention layers to obtain multi-view-aware features $\{\boldsymbol{F}_i\}_{i=1}^N$, $\boldsymbol{F}^i \in \mathbb{R}^{\frac{H}{s} \times \frac{W}{s} \times C}$, where $C$ is the feature dimension. More specifically, cross-attention is performed between each reference view and other views. When more than two images ($N > 2$) are given as input, we perform cross-attention between each reference view and its top-2 nearest neighboring views, which are selected based on their camera position distances to the reference view. This makes the computation tractable while maintaining cross-view interactions.

**Feature Matching**. We encode the feature matching information across different views with the plane-sweep stereo approach (Collins, 1996; Xu et al., 2023). More specifically, for each view $i$, we first uniformly sample $D$ depth candidates $\{d_m\}_{m=1}^D$ from the near and far depth ranges and then warp the feature $\boldsymbol{F}^j$ of view $j$ to view $i$ with the camera projection matrix and each depth candidate $d_m$. Then we obtain $D$ warped features $\{\boldsymbol{F}_{d_m}^{j \to i}\}_{m=1}^D$ that correspond to feature $\boldsymbol{F}^i$. We then measure their feature correlations with the dot-product operation (Xu & Zhang, 2020; Chen et al., 2024). The cost volume $\boldsymbol{C}_i \in \mathbb{R}^{\frac{H}{s} \times \frac{W}{s} \times D}$ for image $i$ is obtained by stacking all correlations. Accordingly, we obtain cost volumes $\{\boldsymbol{C}_i\}_{i=1}^N$ for all input images $\{\boldsymbol{I}_i\}_{i=1}^N$. For more than two input views, similar to the strategy in cross-view attention computation, we select the top-2 nearest views for each reference view and compute feature correlations with only the selected views. This enables our cost volume construction to achieve a good speed-accuracy trade-off and scale efficiently to a larger number of input views. The correlation values for the two selected views are combined with averaging.

### 3.2 MONOCULAR DEPTH FEATURE EXTRACTION

Despite the remarkable progress in multi-view feature matching-based depth estimation (Yao et al., 2018; Wang et al., 2022; Xu et al., 2023) and Gaussian splatting (Chen et al., 2024), they inherently suffer from limitations in challenging situations like occlusions, texture-less regions, and reflective surfaces. Thus, we propose to integrate pre-trained monocular depth features into the cost volume representation to handle scenarios that are challenging or impossible to match.

More specifically, we leverage the pre-trained monocular depth backbone from the recent Depth Anything (Yang et al., 2024b) model thanks to its impressive performance on diverse in-the-wild data. The monocular backbone is a ViT (Dosovitskiy et al., 2020; Oquab et al., 2023) model, which has a patch size of 14 and outputs a feature map that is $1/14$ spatial resolution of the original image. We simply bilinearly interpolate the spatial resolution of the monocular features to the same resolution

as the cost volume in Sec. 3.1 and obtain the monocular feature $\boldsymbol{F}_{\mathrm{mono}}^i \in \mathbb{R}^{\frac{H}{s} \times \frac{W}{s} \times C_{\mathrm{mono}}}$ for input image $\boldsymbol{I}_i$, where $C_{\mathrm{mono}}$ is the dimension of the monocular feature. This process is performed for all input images in parallel and we obtain monocular features $\{\boldsymbol{F}_{\mathrm{mono}}^i \in \mathbb{R}^{\frac{H}{s} \times \frac{W}{s} \times C_{\mathrm{mono}}}\}_{i=1}^N$, which are subsequently used for per-view depth map estimations.

### 3.3 FEATURE FUSION AND DEPTH REGRESSION

To achieve robust and multi-view consistent depth predictions, we combine the monocular feature $\boldsymbol{F}_{\mathrm{mono}}^i \in \mathbb{R}^{\frac{H}{s} \times \frac{W}{s} \times C_{\mathrm{mono}}}$ and cost volume $\boldsymbol{C}_i \in \mathbb{R}^{\frac{H}{s} \times \frac{W}{s} \times D}$ via simple concatenation in the channel dimension. A subsequent 2D U-Net (Rombach et al., 2022; Ronneberger et al., 2015) is used to regress depth from the concatenated monocular features and cost volumes. This process is performed for all the input images in parallel and for each image, it outputs a tensor of shape $\frac{H}{s} \times \frac{W}{s} \times D$, where $D$ is the number of depth candidates. We then normalize the $D$ dimension with the softmax operation and perform a weighted average of all depth candidates to obtain the depth output.

We also apply a hierarchical matching (Gu et al., 2020) architecture where an additional refinement step at $2\times$ higher feature resolution is employed to improve the performance further. More specifically, based on the coarse depth prediction, we perform a correspondence search on the $2\times$ higher feature maps within the neighbors of the $2\times$ upsampled coarse depth prediction. Since we already have a coarse depth prediction, we only need to search a smaller range at the higher resolution, and thus, we construct a smaller cost volume compared to the coarse stage. Such a 2-scale hierarchical architecture not only leads to improved efficiency since most computation is spent on low resolution, but also leads to better results thanks to the use of higher-resolution features (Gu et al., 2020). Similar feature fusion and depth regression procedure is used to get higher-resolution depth predictions, which are subsequently upsampled to the full resolution with a learned upsampler (Ranftl et al., 2021).

### 3.4 GAUSSIAN PARAMETER PREDICTION

For the task of 3D Gaussian splatting, we directly unproject the per-pixel depth maps to 3D with the camera parameters as the Gaussian centers $\boldsymbol{\mu}_j$. We append an additional lightweight network to predict other Gaussian parameters $\alpha_j, \boldsymbol{\Sigma}_j, \boldsymbol{c}_j$, which are opacity, covariance, and color, respectively. With all the predicted 3D Gaussians, we can render novel view images with the Gaussian splatting operation (Kerbl et al., 2023).

### 3.5 TRAINING LOSS

We study the properties of our proposed model on two tasks: depth estimation and novel view synthesis with 3D Gaussian splatting (Kerbl et al., 2023). The loss functions are introduced below.

**Depth estimation**. We train our depth model (without the Gaussian splatting head) with $\ell_1$ loss and gradient loss between the inverse depths of prediction and ground truth:

$$L_{\mathrm{depth}} = \alpha \cdot |\boldsymbol{D}_{\mathrm{pred}} - \boldsymbol{D}_{\mathrm{gt}}| + \beta \cdot (|\partial_x \boldsymbol{D}_{\mathrm{pred}} - \partial_x \boldsymbol{D}_{\mathrm{gt}}| + |\partial_y \boldsymbol{D}_{\mathrm{pred}} - \partial_y \boldsymbol{D}_{\mathrm{gt}}|), \qquad (1)$$

where $\partial_x$ and $\partial_y$ denotes the gradients on the $x$ and $y$ directions, respectively. Following Uni-Match (Xu et al., 2023), we use $\alpha = 20$ and $\beta = 20$.

**View synthesis**. We train our full model with a combination of mean squared error (MSE) and LPIPS (Zhang et al., 2018) losses between rendered and ground truth image colors:

$$L_{\mathrm{gs}} = \sum_{m=1}^M \left( \mathrm{MSE}(I_{\mathrm{render}}^m, I_{\mathrm{gt}}^m) + \lambda \cdot \mathrm{LPIPS}(I_{\mathrm{render}}^m, I_{\mathrm{gt}}^m) \right), \qquad (2)$$

where $M$ is the number of novel views to render in a single forward pass. The LPIPS loss weight $\lambda$ is set to 0.05 following MVSplat (Chen et al., 2024).

## 4 EXPERIMENTS

**Implementation Details**. We implement our method in PyTorch (Paszke et al., 2019) and optimize our model with the AdamW (Loshchilov & Hutter, 2017) optimizer and cosine learning rate learning

Table 1: **DepthSplat model variants**. We explore different monocular backbones and different multi-view models (1-scale and 2-scale features for hierarchical matching as described in Sec. 3.3), where both larger monocular backbones and 2-scale hierarchical models lead to consistently improved performance for both depth estimation and view synthesis tasks.

| Monocular | Multi-View | Depth (TartanAir) | | 3DGS (RealEstate10K) | | | Param |
| | | Abs Rel ↓ | $\delta_1$ ↑ | PSNR ↑ | SSIM ↑ | LPIPS ↓ | (M) |
|---|---|---|---|---|---|---|---|
| ViT-S | 1-scale | 8.46 | 93.02 | 26.76 | 0.877 | 0.123 | 37 |
| ViT-B | 1-scale | 6.94 | 94.46 | 27.09 | 0.881 | 0.119 | 113 |
| ViT-L | 1-scale | **6.07** | **95.52** | **27.34** | **0.885** | **0.118** | 354 |
| ViT-S | 2-scale | 7.01 | 94.56 | 26.96 | 0.880 | 0.122 | 40 |
| ViT-B | 2-scale | 6.22 | 95.31 | 27.27 | 0.885 | 0.120 | 117 |
| ViT-L | 2-scale | **5.57** | **96.07** | **27.44** | **0.887** | **0.119** | 360 |

Table 2: **Ablations**. We evaluate the contribution of the monocular feature branch and the cost volume branch, as well as different monocular features. Our results indicate that the monocular feature and cost volume are complementary, with large performance drops when removing either one. The pre-trained Depth Anything network weights achieve overall the best performance.

| Module | Method | Depth (TartanAir) | | 3DGS (RealEstate10K) | | |
| | | Abs Rel ↓ | $\delta_1$ ↑ | PSNR ↑ | SSIM ↑ | LPIPS ↓ |
|---|---|---|---|---|---|---|
| Components | full | **8.46** | **93.02** | **26.76** | **0.877** | **0.123** |
| | w/o mono feature | 12.25 | 88.00 | 26.27 | 0.866 | 0.130 |
| | w/o cost volume | 11.34 | 90.02 | 23.09 | 0.761 | 0.187 |
| | single branch | 11.26 | 90.84 | 25.99 | 0.858 | 0.134 |
| Monocular features | ConvNeXt-T | 10.50 | 91.13 | 26.28 | 0.867 | 0.130 |
| | Midas | 9.53 | 91.61 | 26.40 | 0.869 | 0.129 |
| | DINO v2 | 8.93 | 92.49 | 26.68 | 0.874 | 0.125 |
| | Depth Anything v1 | **8.38** | **93.23** | 26.70 | 0.875 | 0.125 |
| | Depth Anything v2 | 8.46 | 93.02 | **26.76** | **0.877** | **0.123** |

rate schedule with warm up in the first $5\%$ of the total training iterations. We adopt the xFormers (Lefaudeux et al., 2022) library for our monocular ViT backbone implementation. We use a lower learning rate $2 \times 10^{-6}$ for the pre-trained monocular backbone, and other remaining layers use a learning rate of $2 \times 10^{-4}$. The feature downsampling factor $s$ in our multi-view branch (Sec. 3.1) is chosen based on the image resolution. More specifically, for experiments on the $256 \times 256$ resolution RealEstate10K (Zhou et al., 2018) dataset, we choose $s = 4$. For higher resolution datasets (*e.g.*, TartanAir (Wang et al., 2020), ScanNet (Dai et al., 2017), KITTI (Geiger et al., 2013) and DL3DV (Ling et al., 2023)), we choose $s = 8$. Our hierarchical matching models in Sec. 3.3 use 2-scale features, *i.e.*, $1/8$ and $1/4$, or $1/4$ and $1/2$ resolutions.

**Training Details**. For depth experiments, we mainly follow the setup of UniMatch (Xu et al., 2023) for fair comparisons. More specifically, we train our depth model on 4 GH200 GPUs for 100K iterations with a total batch size of 32. For Gaussian splatting experiments on RealEstate10K (Zhou et al., 2018), we use the same training and testing splits of pixelSplat (Charatan et al., 2024b) and MVSplat (Chen et al., 2024), and train our model on 8 GH200 GPUs for 150K iterations with a total batch size of 32, which takes about 1 day. For experiments on the DL3DV (Ling et al., 2023) dataset, we evaluate on the official benchmark split with 140 scenes, while other remaining 9896 scenes are used for training. We fine-tune our RealEstate10K pre-trained model on 4 A100 GPUs for 100K iterations with a total batch size of 4, where the number of input views is randomly sampled from 2 to 6. We evaluate the model's performance on different number of input views (2, 4, 6). Our code and training scripts will be made publicly available to ease reproducibility.

## 4.1 MODEL VARIANTS

We first study several different model variants for both depth estimation and view synthesis tasks. In particular, we explore different monocular backbones (Yang et al., 2024b) (ViT-S, ViT-B, ViT-L) and different multi-view models (1-scale and 2-scale). We conduct depth experiments on the large-scale

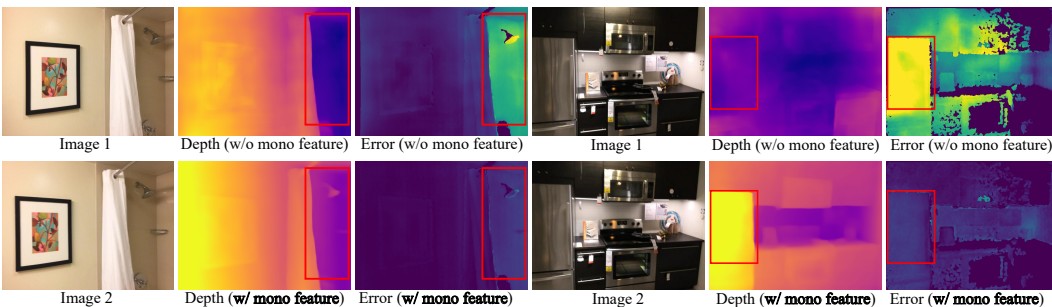

Figure 3: **Effect of monocular features for depth on ScanNet**. The monocular feature greatly improves challenging situations like texture-less regions (*e.g.*, the wall in the first example) and reflective surfaces (*e.g.*, the refrigerator in the second example).

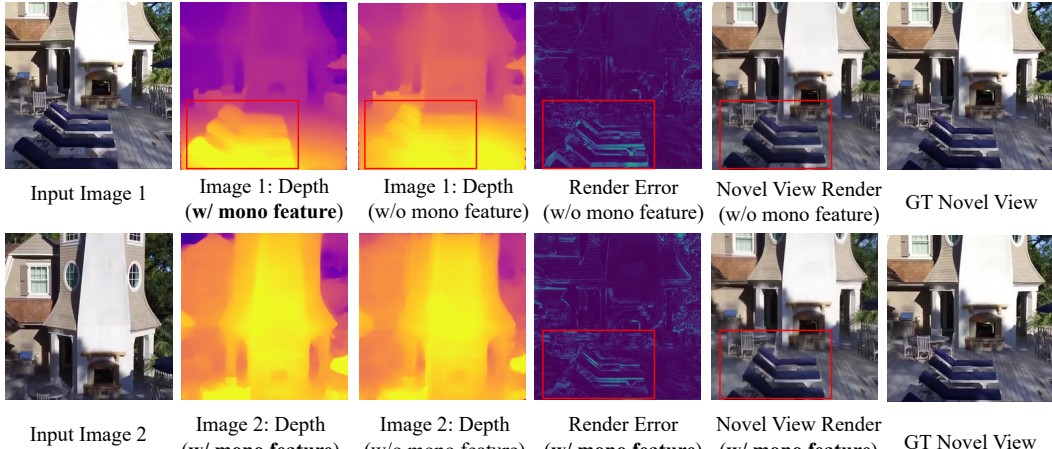

Figure 4: **Effect of monocular features for 3DGS on RealEstate10K**. Without monocular features, the model struggles at predicting reliable depth for pixels that are not able to find correspondences (*e.g.*, the lounger chair highlighted with the read rectangle), which subsequently causes misalignment in the rendered image due to the incorrect geometry.

TartanAir (Wang et al., 2020) synthetic dataset, which features both indoor and outdoor scenes and has perfect ground truth depth. The Gaussian splatting experiments are conducted on the standard RealEstate10K (Zhou et al., 2018) dataset. Following community standards, we report the depth evaluation metrics (Eigen et al., 2014) of Abs Rel (relative $\ell_1$ error) and $\delta_1$ (percentage of correctly estimated pixels within a threshold) and novel view synthesis metrics (Kerbl et al., 2023) of PSNR, SSIM, and LPIPS. The results in Table 1 demonstrate that both larger monocular backbones and 2-scale hierarchical models lead to consistently improved performance for both tasks.

From Table 1, we can also see that better depth network architecture leads to improved view synthesis. In Appendix A.1, we conduct additional experiments to study the effect of different initializations for the depth network to the view synthesis performance. Our results show that a better depth model initialization also contributes to improved rendering quality. In summary, both better depth network architecture and better depth model initialization lead to improved novel view synthesis results.

## 4.2 ABLATION STUDY AND ANALYSIS

In this section, we study the properties of our key components on the TartanAir dataset (for depth estimation) and RealEstate10K dataset (for view synthesis with Gaussian splatting).

**Monocular Features for Depth Estimation and Gaussian Splatting**. In Table 2, we compare our full model (full) with the model variant where the monocular depth feature branch (with a ViT-S model pre-trained by Depth Anything v2 (Yang et al., 2024b)) is removed (w/o mono feature), leaving only the multi-view branch. We can observe a clear performance drop for both depth and view

Table 3: **Unsupervised Depth Pre-Training with Gaussian Splatting**. We first perform unsupervised pre-training with Gaussian splatting on RealEstate10K and then measure the depth performance on TartanAir, ScanNet and KITTI after fine-tuning for the depth task. Compared to previous supervised pre-trained models Depth Anything (for monocular ViT) and UniMatch (for multi-view Transformer), our new unsupervised pre-training improves performance consistently in all metrics. The improvements are especially significant on the challenging datasets like TartanAir and KITTI.

| Initialization | TartanAir | | ScanNet | | KITTI | |
|---|---|---|---|---|---|---|
| | Abs Rel ↓ | $\delta_1$ ↑ | Abs Rel ↓ | $\delta_1$ ↑ | Abs Rel ↓ | $\delta_1$ ↑ |
| Depth Anything (only mono) | 11.12 | 89.97 | 6.78 | 96.13 | 11.67 | 85.96 |
| UniMatch + Depth Anything (mv & mono) | 10.86 | 90.55 | 6.70 | 96.14 | 11.56 | 87.27 |
| DepthSplat (full model) | **10.20** | **91.10** | **6.60** | **96.27** | **10.68** | **89.92** |

synthesis tasks. In Fig. 3, we visualize the depth predictions and error maps of both models on the ScanNet dataset. The pure multi-view feature matching-based approach struggles a lot at texture-less regions and reflective surfaces. At the same time, our full model achieves reliable results thanks to the powerful prior captured in monocular depth features. We also show the visual comparisons for the Gaussian splatting task in Fig. 4 with two input views. For regions (*e.g.*, the lounge chairs) that only appear in a single image, the pure multi-view method is unable to find correspondences. It thus produces unreliable depth predictions, leading to misalignment in the rendered novel views due to the incorrect geometry.

We also experiment with removing the cost volume (w/o cost volume) in the multi-view branch and observed a significant performance drop. This indicates that obtaining scale- and multi-view consistent predictions with a pure monocular depth backbone is challenging, which constrains achieving high-quality results in downstream tasks.

**Fusion Strategy**. We compare with alternative strategies for fusing the monocular features for multi-view depth estimation. In particular, we compare with MVSFormer (Cao et al., 2022) which constructs the cost volume with monocular features. More specifically, we replace our multi-view feature extractor with a weight-sharing ViT model and use the ViT features to build the cost volume as done in MVSFormer. This leads to a single-branch architecture (single branch in Table 2), unlike our two-branch design where the monocular features are not used to build the cost volume. We can observe that our fusion strategy performs significantly better than the single-branch design, potentially because our two-branch design disentangles feature matching and obtaining monocular priors, which makes the learning task easier.

**Different Monocular Features**. In Table 2, we also evaluate different monocular features, including the ConvNeXt-T (Liu et al., 2022) features used in AFNet (Cheng et al., 2024), and other popular monocular features including Midas (Ranftl et al., 2020) and DINO v2 (Oquab et al., 2023). The pre-trained Depth Anything monocular features achieve overall the best performance.

## 4.3 Unsupervised Depth Pre-Training with Gaussian Splatting

By connecting Gaussian splatting and depth, our DepthSplat provides a way to pre-train the depth model in a fully unsupervised manner. In particular, we first train our full model on the large-scale unlabelled RealEstate10K dataset (contains ∼ 67K Youtube videos) with only the Gaussian splatting rendering loss (Eqn. 2), without any direct supervision on the depth predictions. After pre-training, we take the pre-trained depth model and further fine-tune it to the depth task on the mixed TartanAir and VKITTI2 (Cabon et al., 2020) datasets with ground truth depth supervision.

In Table 3, we evaluate the performance on both in-domain TartanAir test set and the zero-shot generalization on unseen ScanNet and KITTI datasets. We compare with only initializing the monocular backbone with Depth Anything and additionally initializing the multi-view Transformer backbone with UniMatch (Xu et al., 2023). Our approach achieves the best results on all three datasets. It is also interesting to observe that our pre-training contributes more on more challenging datasets (*i.e.*, TartanAir and KITTI which feature complex large-scale scenes unlike ScanNet only contains indoor scenes). We also note that both Depth Anything and UniMatch are trained with ground truth supervision, while our DepthSplat is trained with only photometric losses. Given the

Table 4: **Two-view Depth Estimation on Scan-Net.** Our DepthSplat outperforms all prior methods by clear margins.

| Method | Abs Rel ↓ | RMSE ↓ | RMSE_log ↓ |
|---|---|---|---|
| DeMoN | 0.231 | 0.761 | 0.289 |
| BA-Net | 0.161 | 0.346 | 0.214 |
| DeepV2D | 0.057 | 0.168 | 0.080 |
| NeuralRecon | 0.047 | 0.164 | 0.093 |
| DRO | 0.053 | 0.168 | 0.081 |
| UniMatch | 0.059 | 0.179 | 0.082 |
| DepthSplat (ours) | **0.044** | **0.119** | **0.059** |

Table 5: **Two-view 3DGS on RealEstate10K**. Our DepthSplat achieves the best performance.

| Method | PSNR ↑ | SSIM ↑ | LPIPS ↓ |
|---|---|---|---|
| pixelNeRF | 20.43 | 0.589 | 0.550 |
| GPNR | 24.11 | 0.793 | 0.255 |
| AttnRend | 24.78 | 0.820 | 0.213 |
| MuRF | 26.10 | 0.858 | 0.143 |
| pixelSplat | 25.89 | 0.858 | 0.142 |
| MVSplat | 26.39 | 0.869 | 0.128 |
| DepthSplat (ours) | **27.44** | **0.887** | **0.119** |

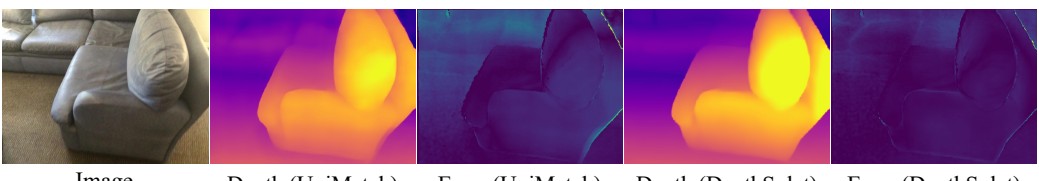

| Image | Depth (UniMatch) | Error (UniMatch) | Depth (DepthSplat) | Error (DepthSplat) |

Figure 5: **Depth Comparison on ScanNet**. Our DepthSplat performs significantly better than UniMatch (Xu et al., 2023) on challenging parts (*e.g.*, edges of the couch and pillows).

increasing popularity of view synthesis (Zheng & Vedaldi, 2024; Weng et al., 2023) and multi-view generative models (Shi et al., 2023; Blattmann et al., 2023), new multi-view datasets (Ling et al., 2023) and models (Voleti et al., 2024) are gradually introduced, our approach provides a way to pre-train depth models on large-scale unlabelled multi-view image datasets. This could potentially further improve the multi-view consistency and robustness of existing depth models (Yang et al., 2024a; Piccinelli et al., 2024), which are usually trained with ground truth depth supervision.

## 4.4 BENCHMARK COMPARISONS

**Comparisons on ScanNet and RealEstate10K**. Table 4 and Table 5 compare the depth and novel view synthesis results on the standard ScanNet and RealEstate10K benchmarks, respectively. For both comparisons, the numbers of input images are two. We can see clearly that our DepthSplat achieves state-of-the-art performance on both datasets for both tasks. The visual comparison with previous best methods is shown in Fig. 5 and Fig. 6, where our method significantly improves the performance on challenging scenarios like texture-less regions and occlusions.

**Comparisons on DL3DV**. To further evaluate the performance on complex real-world scenes, we conduct comparisons with the latest state-of-the-art Gaussian splatting model MVSplat (Chen et al., 2024) on the recently introduced DL3DV dataset (Ling et al., 2023). We also compare the results of different numbers of input views (2, 4 and 6) on this dataset. We fine-tune MVSplat and our RealEstate10K pre-trained models on DL3DV training scenes and report the results on the benchmark test set in Table 6. Our DepthSplat consistently outperforms MVSplat in all metrics. Visual comparisons with MVSplat on the DL3DV dataset are shown in Fig. 7, where MVSplat's depth quality lags behind our DepthSplat due to matching failure, which leads to blurry and distorted view synthesis results. We show more visual comparison results in Appendix A.2. It is also worth noting that our method scales more efficiently to more input views thanks to our lightweight local feature matching approach (Sec 3.1), which is unlike the expensive pair-wise matching used in MVSplat.

We also invite the readers to our supplementary video for the video results on different number of input views (6 and 12), where our DepthSplat is able to reconstruct larger-scale or 360 scenes from more input views, without any optimization.

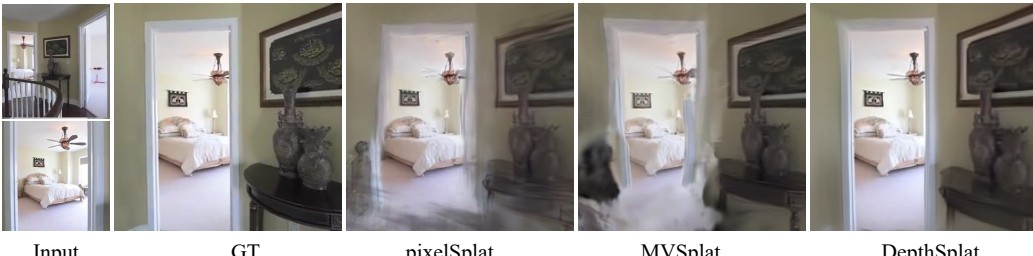

| Input | GT | pixelSplat | MVSplat | DepthSplat |

Figure 6: **Visual Synthesis on RealEstate10K**. Our DepthSplat performs significantly better than pixelSplat (Charatan et al., 2024b) and MVSplat (Chen et al., 2024) in challenging regions.

Table 6: **Comparisons on DL3DV**. Our DepthSplat not only consistently outperforms MVSplat on different number of input views, but also scales more efficiently to more input views.

| Method | #Views | PSNR ↑ | SSIM ↑ | LPIPS ↓ | Time (s) |
|---|---|---|---|---|---|
| MVSplat | 2 | 16.99 | 0.572 | 0.348 | **0.072** |
| DepthSplat | | **17.85** | **0.627** | **0.298** | 0.101 |
| MVSplat | 4 | 20.53 | 0.710 | 0.243 | 0.146 |
| DepthSplat | | **21.28** | **0.744** | **0.208** | **0.124** |
| MVSplat | 6 | 22.85 | 0.772 | 0.193 | 0.263 |
| DepthSplat | | **23.76** | **0.805** | **0.159** | **0.161** |

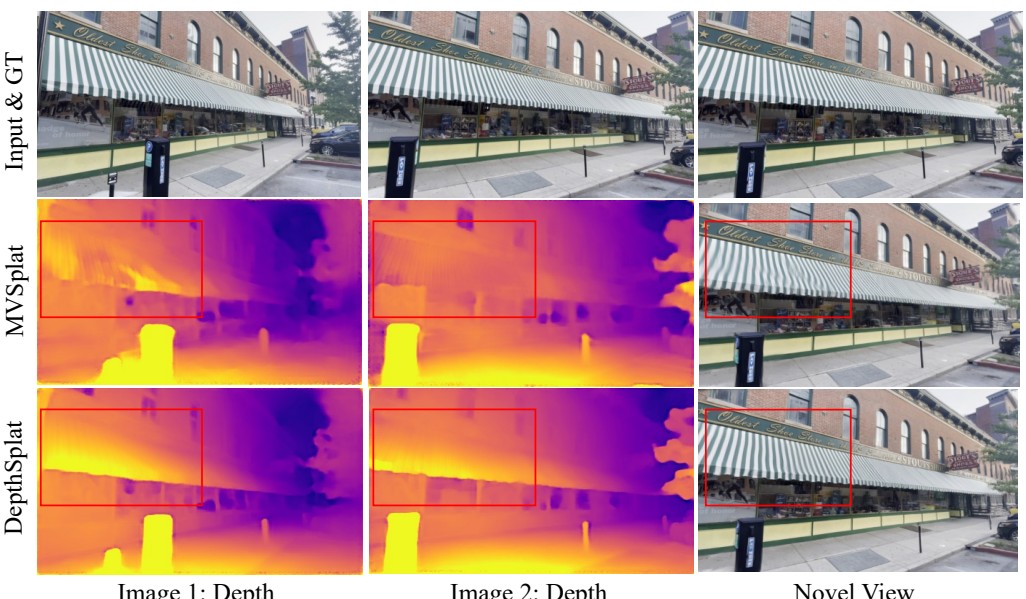

| Image 1: Depth | Image 2: Depth | Novel View |

Figure 7: **Visual Comparisons on DL3DV**. Our DepthSplat performs significantly better than MVSplat (Chen et al., 2024) on regions that hard to match (*e.g.*, repeated patterns).

## 5 CONCLUSION

In this paper, we introduce DepthSplat, a new approach to connecting Gaussian splatting and depth to achieve state-of-the-art results on ScanNet, RealEstate10K and DL3DV datasets for both depth and view synthesis tasks. We also show that our model enables unsupervised pre-training depth with Gaussian splatting rendering loss, providing a way to leverage large-scale unlabelled multi-view image datasets for training more multi-view consistent and robust depth models. Our current model requires camera pose information as input along with the multi-view images, removing this requirement would be exciting future work.

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

# A APPENDIX

## A.1 DEPTH PRE-TRAINING FOR GAUSSIAN SPLATTING

We further study the effect of depth pre-training for Gaussian splatting experiments. Unlike the pre-trained models Depth Anything (Yang et al., 2024b) and UniMatch (Xu et al., 2023), which are trained for either monocular and multi-view features separately, we perform joint training of our full two-branch depth model on the depth datasets. We then compare the results of different initializations to the depth network for the full Gaussian splatting model. We can see from Table 7 that improved depth initialization leads to better view synthesis results.

Table 7: **Depth to Gaussian Splatting Transfer**. We compare different pre-trained network weights for initializing the depth network when training our full DepthSplat model for view synthesis. Compared to using Depth Anything and UniMatch pre-trained monocular and multi-view network weights for initializing the monocular ViT and multi-view Transformer features, our jointly trained 2-branch depth model (full model) performs best on all metrics.

| Initialization | PSNR ↑ | SSIM ↑ | LPIPS ↓ |
|---|---|---|---|
| Depth Anything (only mono) | 26.59 | 0.874 | 0.1256 |
| UniMatch + Depth Anything (mv & mono) | 26.76 | 0.877 | 0.1234 |
| DepthSplat (full model) | **26.81** | **0.878** | **0.1225** |

## A.2 MORE VISUAL COMPARISONS ON DL3DV

We show more visual comparison results on DL3DV in Fig. 8 with different number of input views, where our DepthSplat consistently outperforms MVSplat in challenging regions.

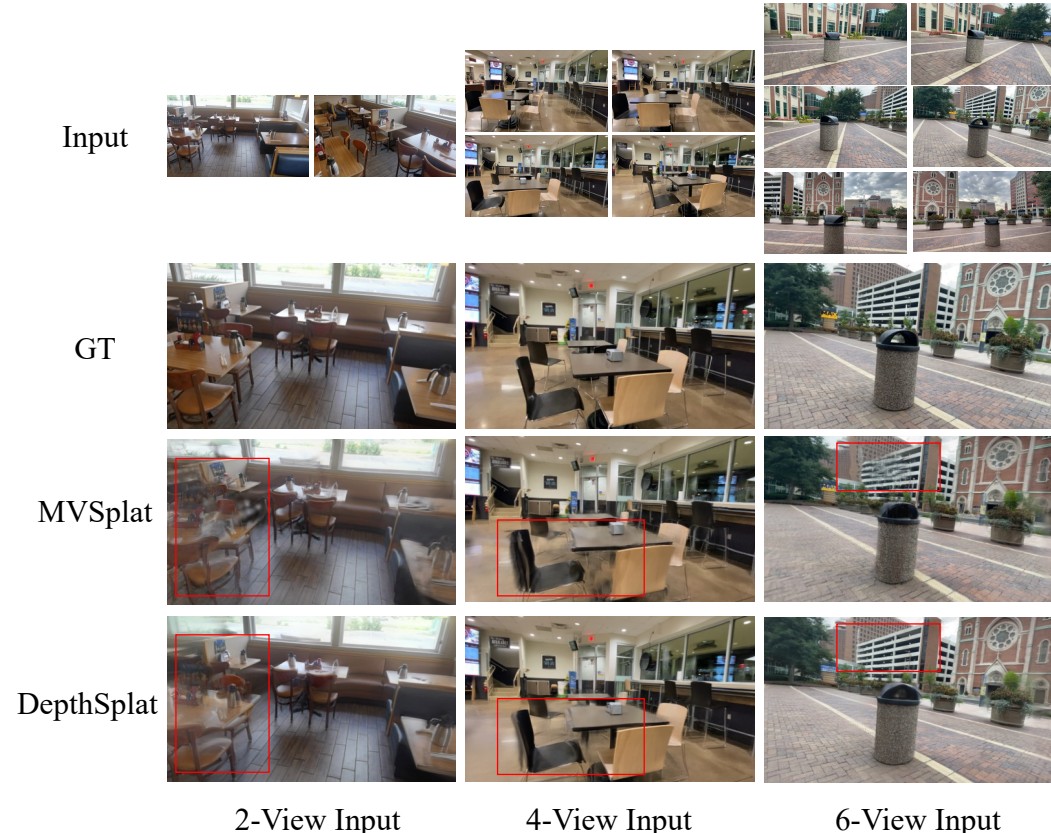

Figure 8: **Different number of input views on DL3DV**. Our DepthSplat performs consistently better than MVSplat (Chen et al., 2024).

