# OpenReview forum: "DepthSplat: Connecting Gaussian Splatting and Depth"
_ICLR.cc/2025/Conference — ICLR 2025 Conference Withdrawn Submission_

### Official Review · Reviewer_6dWD · 2024-10-27

**Soundness:** 3
**Presentation:** 2
**Contribution:** 3
**Rating:** 5
**Confidence:** 4

**Summary:**

This paper introduces a model named DepthSplat, which integrates Gaussian splatting with depth estimation.
The core of the pipeline is a 2D U-Net module that fuses multi-view depth information (from the cost volume) with monocular depth (derived from a pre-trained model, such as Depth Anything).
Experiments conducted on both depth estimation and novel view synthesis tasks demonstrate that the proposed method achieves competitive results in both areas.

**Strengths:**

S1 - The task addressed in this paper is meaningful, and its insights are interesting.

S2 - This paper contributes a robust multi-view depth model by leveraging pre-trained monocular depth features, resulting in high-quality feed-forward 3D Gaussian splatting reconstructions.

S3 - The study demonstrates that Gaussian splatting can serve as an unsupervised pre-training objective for learning powerful depth models from large-scale unlabeled datasets, which is valuable for large-scale pre-trained models.

S4 - Experimental results show that both depth estimation (Table 4) and reconstruction outcomes (Table 5) are significantly improved compared to the tested baselines.

**Weaknesses:**

W1 - The experiment in Section 4.3 appears inconclusive, as it does not convincingly support the claim that "Gaussian splatting can serve as an unsupervised pre-training objective for learning robust depth models from large-scale unlabeled datasets."

W2 - The paper bears similarities to Transplat[1], which also uses monocular depth priors to enhance Gaussian splatting reconstruction. What are the main differences between these two approaches?

W3 - Additional details about DepthSplat, such as its inference speed and memory consumption compared to Mvsplat[2], would be valuable.

[1]Zhang C, Zou Y, Li Z, et al. Transplat: Generalizable 3d gaussian splatting from sparse multi-view images with transformers[J]. arXiv preprint arXiv:2408.13770, 2024.

[2]Chen Y, Xu H, Zheng C, et al. Mvsplat: Efficient 3d gaussian splatting from sparse multi-view images[J]. arXiv preprint arXiv:2403.14627, 2024.

**Questions:**

Q1 - In the experiment in Section 4.3, it is unclear whether DepthAnything and UniMatch were fine-tuned on the test datasets, as this information is not clearly indicated in Lines 425-431.

Q2 - Table 3 appears inconclusive since these three methods utilize different architectures, making it difficult to directly assess the advantage of Gaussian splatting for pre-training. An additional comparison showing DepthSplat with and without pre-training on RE10k could be informative.

Q3 - More details on inference speed and memory cost across different configurations (e.g., depth model, hierarchical architecture) would enhance understanding.

Q4 - What are the key differences between DepthSplat and Transplat?

---

### Official Review · Reviewer_8Qu4 · 2024-11-03

**Soundness:** 3
**Presentation:** 3
**Contribution:** 3
**Rating:** 5
**Confidence:** 4

**Summary:**

This paper presents DepthSplat, which connects Gaussian splatting and depth estimation. Specifically, DepthSplat exploits the complementary nature of sparse view feed-forward 3DGS and robust single/multi-view depth estimation to enhance the performance in both tasks.
For 3DGS: DepthSplat leverages pre-trained monocular depth features (DINOv2) to improve the ability of generalizable of 3D Gaussian reconstructions.
For depth estimation: This paper provides an unsupervised method for pre-training depth prediction models, achieving superior results compared to training from scratch.

**Strengths:**

1. DepthSplat combines the advantages of 3DGS reconstruction and depth estimation to enhance the performance in both tasks. I think this topic is innovative.
2. DepthSplat enhance the depth quality of low-textured areas and reflective surfaces by employing a pre-trained monocular depth model.
3. DepthSplat achieves state-of-the-art results on both tasks.

**Weaknesses:**

1. Please answer the difference between Transplat[1] and DepthSplat. it seems that Transplat also improves the quality of reconstruction by using DINO features.
2. It seems similar to MVSFormer[2] when training the depth model, may I ask the difference between the two methods?

[1] Zhang C, Zou Y, Li Z, et al. Transplat: Generalizable 3d gaussian splatting from sparse multi-view images with transformers[J]. arXiv preprint arXiv:2408.13770, 2024.
[2] Cao C, Ren X, Fu Y. MVSFormer: Multi-view stereo by learning robust image features and temperature-based depth[J]. arXiv preprint arXiv:2208.02541, 2022.

**Questions:**

1. I have some doubts about the training process of DepthSplat. Is the model for predicting gaussian parameters and the model for predicting depth trained separately?
2. Are the depth and 3DGS results in Table 1 evaluated using the same model?
3. Can you provide metrics for DepthSplat pre-trained models using only photimetric loss in Table 3?


If the author can answer my question, I am willing to change my score.

---

### Official Review · Reviewer_ftLe · 2024-11-04

**Soundness:** 3
**Presentation:** 3
**Contribution:** 2
**Rating:** 5
**Confidence:** 4

**Summary:**

The paper introduces DepthSplat, a novel model that integrates 3D Gaussian splatting (3DGS) with single- and multi-view depth estimation, thereby enhancing the performance of both tasks. By leveraging pre-trained monocular depth features, the model produces robust multi-view depth predictions that improve 3DGS outputs. This framework enables pre-training depth model from large scale unlabelled dataset without the ground truth depths. Through this approach, the model achieves state-of-the-art performance in both multi-view depth estimation and novel view synthesis tasks.

**Strengths:**

- The paper leverages pre-trained monocular depth models effectively to achieve strong and superior performance in both depth estimation and novel view rendering.
- DepthSplat achieves efficient memory usage and computational scalability, allowing it to handle large-scale scenes and even produce 360° views. This is a substantial improvement for feed-forward 3DGS models.

**Weaknesses:**

1. Can the authors analyze how DepthSplat’s depth estimation performance varies with different numbers of input views? This would provide a better understanding of its robustness in sparse-view settings. Moreover, I would like to see how it performs with varying number of inputs in terms of PSNR, SSIM and LPIPS, when evaluated on RealEstate10K.
2.  I would like to see the efficiency comparison (memory consumption) to MVSplat [1], with varying the number of input views. Also, I am quite confused by the speed comparison in Table 6. If I understood correctly, the proposed method shares similar architecture to MVSplat, besides it adds monocular feature extraction part, which should increase the computation. Why does it infer faster than MVSplat?
3. For N-view evaluation, It seems to show different observation from [2, 3]. In [2, 3], MVSplat's synthesized image qualities drastically drop when the number of views increase. However, in the Table 6, it seems to increase. What is the difference that led to such differing results? If the number of gaussians increase, unless a module to control the number of Gaussians is proposed [2, 3] I was assuming that the results should not improve significantly. Is it because the authors use top-2 nearest views only for cost computation? However, even so, this does not explain the apparent improvements despite redundant 3D Gaussians depicting the same object surface.
4. The paper does not provide a detailed description of the architecture for the 3D Gaussian prediction module. This component is central to the view synthesis pipeline, so more information on its design would improve clarity and reproducibility.
5. The paper lacks results on the depth estimation performance after unsupervised pre-training alone, without additional fine-tuning. Having these results would help evaluate the standalone effectiveness of the unsupervised pre-training step for depth tasks.
6. As noted in the related works section, DepthSplat effectively augments multi-view cost volumes with pre-trained monocular depth estimation features. However, its approach closely resembles existing models that fuse monocular and multi-view depth estimations, such as those presented in [4, 5], particularly in how they handle the fusion of these estimations, aside from DepthSplat's use of pre-trained models. The biggest differences may be either concatenation or cross-attention between monocular branch and multi-view branch, and the use of pre-trained large model weights. To me, the novelty and effectiveness are questionable, since without pretrained weights, the performance would drop significantly, and the fusing part is highly similar to previous works. Providing more justification on how the integration of pre-trained models constitutes a novel contribution would help strengthen the paper.

[1] Chen, Yuedong, et al. "Mvsplat: Efficient 3d gaussian splatting from sparse multi-view images." arXiv preprint arXiv:2403.14627 (2024).

[2] Fei, Xin, et al. "PixelGaussian: Generalizable 3D Gaussian Reconstruction from Arbitrary Views." arXiv preprint arXiv:2410.18979 (2024).

[3] Wang, Yunsong, et al. "FreeSplat: Generalizable 3D Gaussian Splatting Towards Free-View Synthesis of Indoor Scenes." arXiv preprint arXiv:2405.17958 (2024).

[4] Li, Rui, et al. "Learning to fuse monocular and multi-view cues for multi-frame depth estimation in dynamic scenes." Proceedings of the IEEE/CVF Conference on Computer Vision and Pattern Recognition. 2023.

[5] Cheng, Junda, et al. "Adaptive fusion of single-view and multi-view depth for autonomous driving." Proceedings of the IEEE/CVF Conference on Computer Vision and Pattern Recognition. 2024.

**Questions:**

1. Pre-trained models like DUSt3R[1] and MASt3R[2], which perform well on both monocular and multi-view tasks, might serve as viable alternatives to DepthSplat's dedicated monocular branch. Have the authors considered using pre-trained models like DUSt3R or MASt3R, which could potentially simplify the architecture? What benefits does DepthSplat’s separate monocular branch provide in comparison?
2. How well does DepthSplat generalize across datasets, such as when a model trained on RealEstate10K is tested on DL3DV? Would it maintain performance in depth estimation and novel view synthesis on previously unseen datasets?
3. Would it be possible if DepthSplat's depth estimation result is compared with 3DGS works like MVSplat or PixelSplat? This comparison would be very interesting to see.

[1] Wang, Shuzhe, et al. "Dust3r: Geometric 3d vision made easy." Proceedings of the IEEE/CVF Conference on Computer Vision and Pattern Recognition. 2024.

[2] Leroy, Vincent, Yohann Cabon, and Jérôme Revaud. "Grounding Image Matching in 3D with MASt3R." arXiv preprint arXiv:2406.09756 (2024).

**Details Of Ethics Concerns:**

There are no concerns.

---

### Official Review · Reviewer_1cTc · 2024-11-04

**Soundness:** 4
**Presentation:** 3
**Contribution:** 2
**Rating:** 5
**Confidence:** 4

**Summary:**

The paper presents DepthSplat, a novel approach that connects 3D Gaussian Splatting (3DGS) with multi-view and monocular depth estimation to enhance both depth prediction and generalized novel view synthesis. Leveraging pre-trained monocular depth features from the Monocular Depth Estimation(MDE) model, DepthSplat improves the robustness of depth estimation, especially in challenging scenes with occlusions, texture-less regions, and reflective surfaces, which are typically problematic for multi-view methods alone. By integrating monocular features into a dual-branch model architecture DepthSplat achieves both multi-view consistency and high-quality reconstruction. Ablation studies show that monocular features are crucial for best performance, especially in difficult areas, and that DepthSplat's dual-branch design outperforms single-branch alternatives by simplifying the learning process. As a result, the presented DepthSplat archives state-of-the-art performance in multiple datasets.

**Strengths:**

- The overall paper is well-written and easy to understand, where most of the architectural designs and training details are also specifically mentioned, making the re-implementation of the paper easy.
- From the perspective of generalized feed-forward 3DGS, I believe that existing works still struggle to extend to multiple views whereas DepthSplat enables up to novel view synthesis with 6 views.
- Building up from a pre-trained Monocular Depth Estimation (MDE) Module, DepthSplat improves the performance of both Novel View Synthesis and Depth Estimation.
- The choice of the backbone MDE network and the components of the architecture is discussed through well-conducted ablation studies.

**Weaknesses:**

Although the proposed contributions of DepthSplat are valuable, I believe that not all of the mentioned contributions are shown in the paper. Specifically,

- Some of the architectures are not mentioned explicitly in the paper, such as how the Gaussian parameters are estimated.
- Although the authors mention that the pre-training of the network only with the Gaussian Splatting rendering loss is a way to pre-train the depth model in a fully unsupervised manner, there is no explanation or ablation of how this pre-training improves the depth prediction performance.

**Questions:**

From the initial manuscript, I have some concerns and questions about the proposed method. With the questions properly addressed, I am eager to raise my ratings.

1. About the pre-trained Monocular Depth Estimation Module the authors are leveraging to initialize the weights of the monocular backbone, I am a bit confused if the authors are utilizing DepthAnythingv1[1] or DepthAnythingv2[2]. In most of the parts, the model is only specified as DepthAnything which is not clear which version of the model the authors are utilizing. Specifically, in Section 4.2, it is mentioned that DepthAnythingv2 is utilized but in Table 2, v1 is best in depth prediction and v2 is best in novel view synthesis and the authors simply mention that DepthAnything shows the best performance. I would suggest the authors to explicitly mention the version of the Depth Anything network they are utilizing as it is a bit confusing.

2. What is the architectural design for the Gaussian parameter prediction? It is only mentioned as “We append an additional lightweight network to
predict other Gaussian parameters” but I can’t not find any additional information about the lightweight network in the main paper or the appendix.

3. How does the pre-training stage only with the Gaussian Splatting rendering loss improve the performance of depth estimation? Specifically, I would like to see the comparison between the Depth prediction branch fine-tuned directly with the depth labels and with the pre-training stage.

4. As the authors claim that DepthSplat can achieve multi-view consistent depths, are there any evaluations that can show the superiority of DepthSplat on multi-view consistency? For example, a recent work called FreeSplat[3] mentions that when using per-pixel Gaussians, due to the slight inconsistent position of the Gaussians of corresponding pixels, this leads to quality degradation and proposes a Gaussian fusion method. I believe that with multi-view consistent depth, this type of additional method is not strictly required. I would appreciate if the authors could explain how the multi-view consistency can be understood or evaluated.

---

[1] Yang, Lihe, et al. "Depth anything: Unleashing the power of large-scale unlabeled data." Proceedings of the IEEE/CVF Conference on Computer Vision and Pattern Recognition. 2024.

[2] Yang, Lihe, et al. "Depth Anything V2." arXiv preprint arXiv:2406.09414 (2024).

[3] Wang, Yunsong, et al. "FreeSplat: Generalizable 3D Gaussian Splatting Towards Free-View Synthesis of Indoor Scenes." *arXiv preprint arXiv:2405.17958* (2024).

**Details Of Ethics Concerns:**

There are no concerns.

---

### Official Review · Reviewer_4tRN · 2024-11-04

**Soundness:** 3
**Presentation:** 3
**Contribution:** 3
**Rating:** 5
**Confidence:** 4

**Summary:**

This paper introduces DepthSplat, a framework that bridges Gaussian splatting and depth estimation to improve both tasks. By incorporating pre-trained monocular depth features, the proposed method enhances 3D Gaussian splatting for novel view synthesis and enables unsupervised pre-training of depth models on unlabeled datasets. Extensive experiments demonstrate that DepthSplat achieves state-of-the-art performance in both depth estimation and view synthesis.

**Strengths:**

1. Unified Architecture: The architecture of DepthSplat seamlessly integrates depth estimation and 3D Gaussian splatting, allowing for efficient cross-task knowledge transfer. This integration addresses the limitations of each technique while enhancing their complementary strengths.

2. Comprehensive Experiments: This paper validates its approach through extensive experiments, showcasing the model's robustness and generalizability.

**Weaknesses:**

Although the proposed method achieves impressive performance, its technical contribution is relatively weak.
The authors claim two main contributions: integrating pretrained monocular depth features into the multiview depth model and demonstrating the capability of Gaussian Splatting to serve as an objective for unsupervised depth learning.

First, the approach to fuse multiview features and monocular features is straightforward, and the idea has been explored in previous works, such as [1, 2].

Second, while using Gaussian Splatting for unsupervised depth learning is interesting, it is not particularly innovative, as view-synthesis based approaches have been the mainstream for unsupervised depth estimation. Neural rendering methods, such as NeRF, have already been integrated into unsupervised depth estimation in previous works [3].

Finally, the feedforward framework proposed in this paper is largely similar to previous works, such as Pixelsplat, with the main difference being the architecture of the depth estimation network.

[1] MultiView Depth Estimation by Fusing Single-View Depth Probability With MultiView Geometry. CVPR 2022

[2] Learning To Fuse Monocular and MultiView Cues for MultiFrame Depth Estimation in Dynamic Scenes. CVPR 2023

[3] DeLiRa: Self-Supervised Depth, Light, and Radiance Fields. ICCV 2023

**Questions:**

Please see the weaknesses.

---

### Note · Authors · 2024-11-15

I have read and agree with the venue's withdrawal policy on behalf of myself and my co-authors.